# B-1a cells acquire their unique characteristics by bypassing the pre-BCR selection stage

Jason B. Wong[1], Susannah L. Hewitt [1], Lynn M. Heltemes-Harris[2], Malay Mandal [3], Kristen Johnson[1], Klaus Rajewsky [4], Sergei B. Koralov[1], Marcus R. Clark [3], Michael A. Farrar [2] & Jane A. Skok [1]*

B-1a cells are long-lived, self-renewing innate-like B cells that predominantly inhabit the peritoneal and pleural cavities. In contrast to conventional B-2 cells, B-1a cells have a receptor repertoire that is biased towards bacterial and self-antigens, promoting a rapid response to infection and clearing of apoptotic cells. Although B-1a cells are known to primarily originate from fetal tissues, the mechanisms by which they arise has been a topic of debate for many years. Here we show that in the fetal liver versus bone marrow environment, reduced IL-7R/ STAT5 levels promote *immunoglobulin kappa* gene recombination at the early pro-B cell stage. As a result, differentiating B cells can directly generate a mature B cell receptor (BCR) and bypass the requirement for a pre-BCR and pairing with surrogate light chain. This 'alternate pathway' of development enables the production of B cells with self-reactive, skewed specificity receptors that are peculiar to the B-1a compartment. Together our findings connect seemingly opposing lineage and selection models of B-1a cell development and explain how these cells acquire their unique properties.

[1] Department of Pathology, New York University School of Medicine, New York University, New York, NY, USA. [2] Department of Laboratory Medicine and Pathology, Center for Immunology, Masonic Cancer Center, University of Minnesota, Minneapolis, MN, USA. [3] Department of Medicine, Section of Rheumatology and Gwen Knapp Center for Lupus and Immunology Research, University of Chicago, Chicago, IL, USA. [4] Max Delbrück Center for Molecular Medicine, 13092 Berlin, Germany. *email: jane.skok@nyumc.org

B-CLL is the most common form of adult leukemia in the western world[1]. In the early 1980s, the T-cell antigen CD5 (Ly-1) was identified on the surface of cancerous B cells in patients with B-CLL[2,3]. This observation led to the search for normal CD5$^+$ B-cell counterparts to potentially determine the cancer cell of origin. As a result of these efforts, CD5$^+$ B cells, otherwise known as B-1a cells, were discovered in mice[4,5]. Further characterization of CD5$^+$ B-1a B cells revealed that they are long-lived, self-renewing cells that predominantly reside in the pleural and peritoneal cavities where they produce natural polyreactive IgM antibodies with a biased, autoreactive repertoire. In contrast to conventional B-2 cells, B-1a cells produce antibodies with reduced junctional diversity and less somatic hypermutation[6]. Furthermore, $Igh$ $V_H$ gene rearrangements favor $V_H$12 segment usage[7], generating antibodies that interact with phosphatidylcholine (PtC), a major lipid in the protective mucus layer of the gastrointestinal tract that is also present in the membranes of diverse bacteria. Thus, the B-1a receptor repertoire is biased toward bacterial and self-antigens, which is important for mounting a rapid immune response to infection and in the clearing of apoptotic cells[8–10]. Because B-1a cells are found in pre-immune mice, they function as an important first line of defense against bacterial pathogens. These characteristics distinguish B-1a cells from conventional B-2 cells, which have a highly diverse receptor repertoire that is important for mediating adaptive immunity.

Although B-1a cells were discovered in the early 1990s, their origin has been hotly debated since, and despite the efforts of numerous labs this remains an unresolved issue. The controversy has mainly been centered on two opposing models, the lineage model and the selection model. The lineage model proposes that a distinct B-1 progenitor cell gives rise to B-1a cells, while the selection model favors the idea that a common B-cell progenitor can acquire a B-1a or a B-2 fate depending on the type of antigen it recognizes[9,11]. Support for the lineage model comes from early reconstitution experiments, which reveal that fetal tissues are much more efficient at generating B-1a cells in irradiated recipient mice than adult bone marrow counterparts[12]. Furthermore, the first wave of B-1a cells was shown to originate in early embryos in an HSC-independent manner[13–17]. However, cellular barcoding experiments demonstrate that a single progenitor cell can give rise to both B-1a and B-2 cells[18] challenging the notion that B-1a cells arise from a distinct lineage. Moreover, the finding that B-1a cells have a restricted and biased receptor repertoire provides support for a selection model[9,19]. Further support for the selection model comes from a study by Graf et al. that made use of a transgenic system to show that swapping B-2 and B-1a-specific B-cell receptors (BCRs) is sufficient to efficiently change a B-2 cell into a B-1a cell in the absence of any lineage constraints. The lineage switch is rapid, induces a proliferative burst, and cells migrate to their normal environments within the pleural and peritoneal cavities[20].

Investigations have also focused on expression of specific genes that influence development. For example, the $Lin28b$/let7 molecular switch influences CD5$^+$ B-1a cell fate[21–23]. The Lin28b and let7 miRNAs, respectively, promote and inhibit the expression of the transcription factor, $Arid3$a, which in turn can drive the development of CD5$^+$ B-1a cells[22,23]. Nonetheless, the BCR repertoire of the resulting cells do not include PtC-specific antibodies that are characteristic of a typical B-1a cell compartment[22]. Thus, enforced expression of $Arid3a$ fails to fully explain how B-1a cells develop. Another transcription factor, BHLHE41 has also been shown to be important in B-1a cell biology[24]. Specifically, cells deficient in this transcription factor lose B-1a cells expressing $V_H$12/$V_K$4 PtC-specific receptors, have impaired BCR signaling, increased proliferation, and apoptosis. BHLHE41 therefore plays an important role in B-1a maintenance by regulating self-renewal and BCR repertoire; however, it is not known whether its forced expression can drive development of these cells.

In the fetus, B-cell development takes place in the liver and moves to the bone marrow after birth. Each stage of development is marked by a particular rearrangement event that drives differentiation forward. These recombination events occur in a stage-specific manner. The first step involves the joining of the $immunoglobulin$ $heavy$-$chain$ ($Igh$) $D_H$ and $J_H$ gene segments within pre-pro-B cells. Rearrangement continues at the pro-B cell stage, where $V_H$-to-$DJ_H$ joining is both initiated and completed. Rearrangement of the $immunoglobulin$ $light$-$chain$ gene loci, $Igk$ or $Igl$, occurs at the subsequent pre-B cell stage of development. $Igh$ and $Igk$ gene rearrangement is separated by a proliferative burst of large pre-B cells that allows individual cells that have successfully rearranged their heavy chain to clonally expand. At the following small pre-B cell stage, each B-cell undergoes a distinct $Igk$ gene recombination event[25]. Ultimately, this results in unique heavy- and light-chain pairs that expand the antigen receptor repertoire. The successful pairing of immunoglobulin heavy chain with surrogate light chain (SLC) forms the pre-B cell receptor (pre-BCR), which is required for expansion of large pre-B cells and subsequent differentiation to the small pre-B cell stage, where $Igk$ recombination occurs. Since SLC pairs poorly with autoreactive heavy chains, the pre-BCR provides a mechanism for negative selection of self-reactive B cells[26,27].

As noted early on, B-1 cell development is quantitatively unaffected in SLC-deficient mice, and in a small fraction of B-cell progenitors in the bone marrow $Igk$ rearrangements occur prior to rearrangements at $Igh$ and independent of SLCs[28–31]. In addition, in the absence of SLC and thus pre-BCR expression, an autoreactive BCR can drive B-cell development efficiently to the stage of the immature B cell, where BCR diversification and counterselection of autoreactivity is achieved through the process of receptor editing. This has led to a model in which early B-cell development is driven by a positive signal from the pre-BCR (in the majority of progenitors) or an autoreactive BCR (in a minority of cells), with the pre-BCR having evolved as a surrogate autoreactive BCR[32,33].

Downstream of pre-BCR signaling, IL-7 receptor (IL-7R) signaling is extinguished at the small pre-B cell stage. This is noteworthy because the IL-7R signaling pathway is responsible for directing the sequential ordering of recombination events, i.e., $heavy$-$chain$ $gene$ followed by $light$-$chain$ $gene$ rearrangement in classical B-cell development[34,35]. IL-7 is a cytokine secreted by stromal cells within the bone marrow where development takes place throughout the postnatal and adult life of the animal. In murine bone marrow, development past the pre-pro-B cell stage stringently requires IL-7R[36,37]. In contrast, B-cell development within the fetal liver can occur independent of IL-7[38,39]. In terms of directing recombination, IL-7 and its downstream signaling component STAT5 have been shown to promote $Igh$ accessibility and recombination[40–42] while actively inhibiting recombination of the $Igk$ locus[34,43]. Activated STAT5 enters the nucleus and forms a complex with PRC2/EZH2, which binds to the intronic enhancer of $Igk$ (iEκ) and induces H3K27me3-mediated repression to inhibit recombination of this locus[44,45]. Strikingly, conditional deletion of STAT5 within pro-B cells results in a robust increase of $Igk$ recombination at the pro-B cell stage[35].

It is known that B-1a cells originate primarily from fetal tissues, however, it remains unclear what pathways drive B-1a cell development and lead to the acquisition of their unique characteristics. In addition, it has been shown that B-1a cells are efficiently generated in $Il7^{-/-}$ and $Il7r^{-/-}$ mice albeit at reduced numbers compared with wild-type[38,39,46,47]. Given our finding that fetal liver (FL) pro-B cells have less active cytoplasmic

pSTAT5 than bone marrow (BM) pro-B cells[48], we asked whether this could influence V(D)J recombination and B-cell development.

Here, we show that low levels of IL-7R/pSTAT5 signaling in the fetal liver environment promote an "alternate pathway" of B-cell development, in which increased *Igk* rearrangement occurs at the pro-B cell stage of development. Productive rearrangement of *Igh* and *Igk* at this stage leads directly to cell surface expression of a mature BCR, bypassing the requirement for SLC and the selection of autoreactive receptors that are characteristic of B-1a cells. Extending earlier work, we demonstrate that while SLC-independent development leads to a significant reduction in B-2 cell numbers, B-1a cells with their characteristic $V_H12$ anti-PtC bias are still efficiently generated. Thus, reduced IL-7R/STAT5 signaling promotes an alternate pathway of development which favors the production of B-1a cells with self-reactive receptors characteristic of this B-cell subset. Together these data connect opposing lineage and selection models of B-1a cell development and explain how these cells acquire their unique properties.

## Results

**Igk recombination is increased in fetal liver pro-B cells.** Fetal liver (FL) and bone marrow (BM)-derived progenitor cells are differentially dependent on IL-7 for development, and we have found that FL-derived pro-B cells have significantly lower levels of the downstream signaling component, phospho-STAT5 compared with their bone marrow-derived pro-B cell counterparts[48]. Thus, we asked whether alterations in phospho-STAT5 levels correlate with altered regulation of *Igk* recombination in these two anatomic locations. To explore this, we analyzed recombination from ex vivo-derived bone marrow (5–6 wk adults) and FL (E17–18) pro-B cells using semiquantitative PCR analysis (Fig. 1a). This assay uses a degenerate Vκ primer with a common reverse primer downstream of Jκ gene segments to measure recombination of a Vκ with each of the four Jκ segments[49]. As shown in Fig. 1a, FL pro-B cells have increased *Igk* recombination relative to BM pro-B cells.

To address the question of altered *Igk* recombination between these two cell types in a more quantitative manner, we made use of a real-time PCR assay that quantifies Jκ1 rearrangement by assessing the retention of germline *Igk*[50]. In this assay, a product cannot be generated after recombination, because the sequence to which the upstream primer binds is lost or inverted during the recombination process. Quantitation of the amplified product is determined as a ratio between the single copy gene β-actin and the germline *Igk* band. Control tail DNA, in which no *Igk* rearrangement occurs, is set at 100%. Consistent with the semiquantitative experiments described above, FL pro-B cells have lower levels of *Igk* germline retention (61%) relative to bone marrow pro-B cells (87%) (Fig. 1b).

To determine the proportion of cells that are actively undergoing recombination at a single-cell level, we used three-dimensional (3-D) immunofluorescent in situ hybridization (FISH) assay that visualizes recombination by the association of γ-H2AX DNA repair foci with the individual alleles of antigen receptor loci[51,52]. In these experiments, we used two *Igk* BAC DNA probes that hybridize to the distal Vκ24 (RP23–101G13) gene region and the Cκ region (RP24–387E13) in combination with an antibody against the phosphorylated form of γ-H2AX (Fig. 1c). We found significantly increased association of γ-H2AX foci with the *Igk* locus in FL versus bone marrow-derived pro-B cells (16.7% versus 3.0%). Consistent with our previous experiments, the frequency of DNA breaks in FL pro-B cells (16.7%) is again at an intermediate level compared with BM pre-B cells

(30%) where *Igk* recombination typically occurs (Fig. 1d). These numbers reflect the combined data from two independent experiments (Supplementary Table 1).

It is known that activated STAT5 forms a complex with PRC2 and binds to the intronic enhancer of *Igk*, iEκ to inhibit *Igk* recombination[44,45,48]. To determine if the increase in early *Igk* rearrangement could be caused by reduced STAT5-mediated repression, we analyzed enrichment of STAT5 binding on iEκ via STAT5-ChIP-qPCR. As shown in Fig. 1e, we found that FL pro-B cells have significantly less STAT5 bound to iEκ compared with BM pro-B cells, and binding levels more closely resemble those in BM pre-B cells where *Igk* recombination normally occurs. Taken together, these findings indicate that in the FL versus BM environment, *Igk* rearrangement occurs at higher levels in pro-B cells due to a decrease in STAT5-mediated repression of the *Igk* locus.

Antigen receptor loci undergo large-scale contraction through chromatin looping, which enables rearrangement between widely dispersed gene segments. Locus contraction is tightly linked both to recombination status and usage of distal gene segments[49,53]. To address the contraction status of the *Igk* locus, we measured the distance separating the distal Vκ24 and Cκ probes in interphase cells. Our analyses indicate that FL pro-B cells are significantly more contracted than wild-type (WT) double-positive (DP) T cells, a cell type in which *Igk* rearrangement does not occur (Fig. 2). Indeed, the level of contraction is comparable with that seen in wild-type bone marrow-derived pre-B cells, where *Igk* gene rearrangement is known to occur.

**Bypass of the pre-BCR checkpoint.** Classical bone marrow B-cell development starts with *Igh* recombination at the pre-pro to pro-B cell stages followed by *Igk/Igl* recombination at the small pre-B cell stage. However, JHT mice that lack J_H segments as well as the *Igh* intronic enhancer, and are therefore unable to undergo *Igh* recombination, but can rearrange *Igk/Igl* independent of *Igh* rearrangement (Fig. 3a)[29,54,55]. This is also true for SLC-deficient mice, and a fraction of B cells undergoing development in the bone marrow of wild-type mice[56]. Given our finding that a significant number of pro-B cells rearrange *Igk* in the FL environment, we reasoned that productive *heavy* and *light-chain gene* rearrangement in pro-B cells could lead to expression of a mature BCR, bypassing the requirement for a pre-BCR. To address this question as a proof of principle, we examined FL B cells from mice that are deficient for a component of SLC, lambda5 (*Igll1*$^{-/-}$), in conjunction with a transgene that has a prearranged functional heavy chain (B1.8). *Igll1*$^{-/-}$ mice display a strong developmental block at the pro-B cell stage and made very few IgM$^+$ cells (0.59% of B cells). In contrast *Igll1*$^{-/-}$, B1.8 and B1.8 mice produce similarly high percentages of IgM + B cells (12 and 13.5%, respectively). Importantly, B1.8 mice have a substantial pre-B cell compartment which is not observed in *Igll1*$^{-/-}$, B1.8 mice despite a similar output of IgM$^+$ B cells which express IgK protein (Fig. 3b). This shows that having a productive heavy- and light chain can bypass the requirement for pairing with SLC to form immature B cells that express the mature BCR. Together these data demonstrate that in the FL, expression of *Igh* and *Igk* at the pro-B cell stage can bypass the developmental block induced by surrogate light-chain deficiency by forming B cells with a mature BCR.

**B-1a cells are efficiently generated in *Igll1*$^{-/-}$ mice.** B-1a cells are primarily generated from fetal tissues, and we have shown that FL-derived B cells frequently undergo early *Igk* recombination allowing cells to bypass the pre-BCR selection stage. Thus, we next asked whether an absence of SLC could impact the generation of B-1a cells[12,23]. Here, we analyzed the B-2 as well as

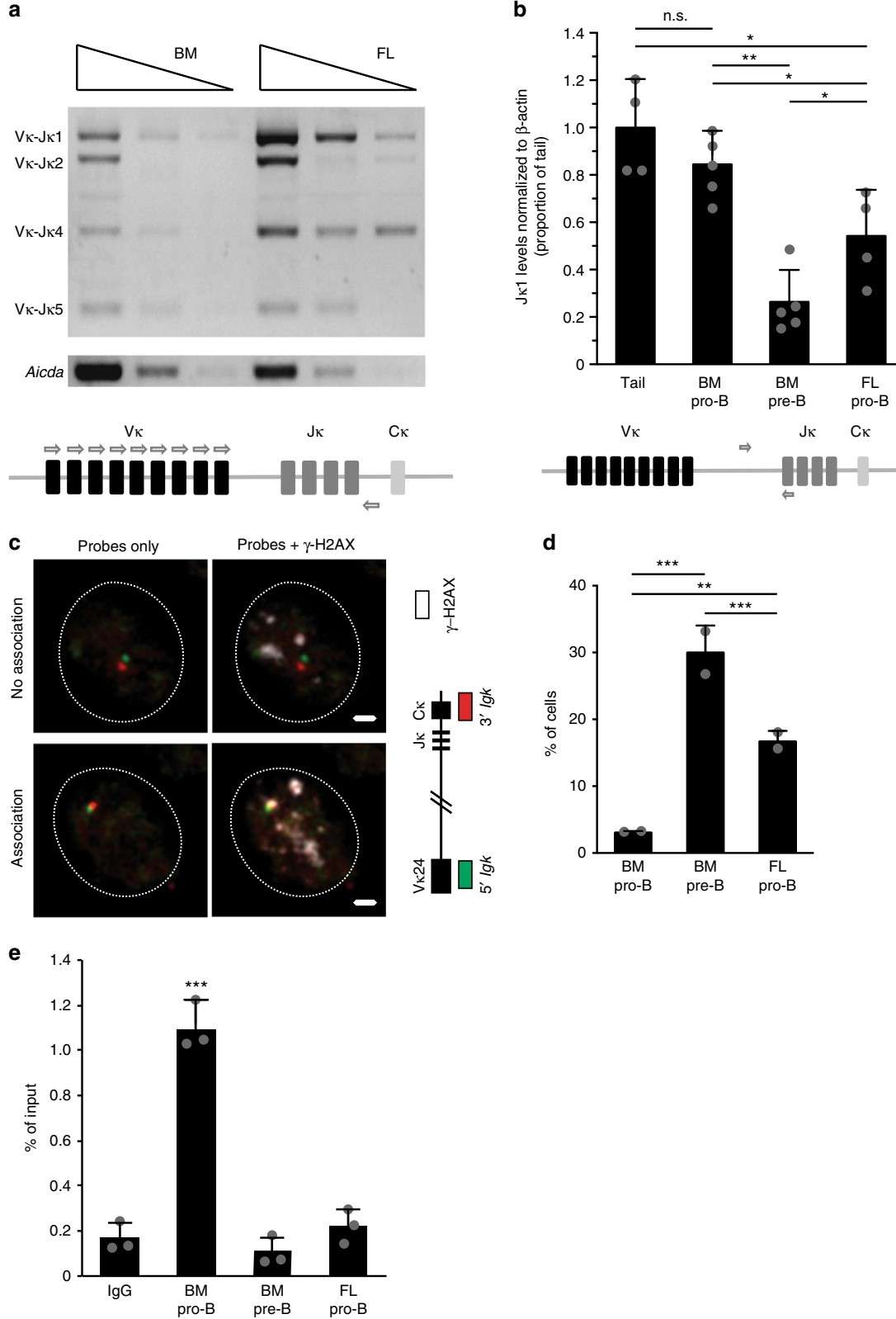

the B-1 compartment, which consists of both B-1a and B-1b cells. B-1b cells are distinct from B-1a cells in that they lack CD5 expression and have a memory function in protecting against bacterial infections[57]. As expected, peritoneal B-2 cells were significantly decreased compared with controls. Furthermore, although B-1b cells were also significantly decreased, the B-1a cell compartment remained intact. This was reflected in cell percentage, as well as total cell numbers (Fig. 4a, b). These data indicate that while B2 and B1b cell development is strongly promoted by SLC, B-1a cells do not depend on a pre-BCR checkpoint. It should be noted that B1.8 transgenic mice could not be used to analyze the B-1a cell compartment in SLC-deficient mice, because the B1.8 pre-rearranged heavy chain does not support B-1a cell development[58].

**Fig. 1** Early *Igk* recombination is increased in the fetal liver versus bone marrow pro-B cells. **a** Semiquantitative PCR performed on ex vivo-derived BM pro-B (CD19+B220+IgM−CD25−CD117+) from 6-week-old mice and FL pro-B (Lin−CD19+B220+IgM−CD2−CD117+) from E17.5 mice to assess rearrangement of Vκ to each functional Jκ. Each lane represents threefold serial dilutions of input DNA. DNA levels were normalized to *Aicda* levels, which is located on the same chromosome as *Igk*. **b** Recombination with Jκ1 was quantified by qPCR using primers specific for the unrearranged germline sequence. DNA is quantified as a ratio between the single copy *β-actin* gene and the *Igk* germline PCR product and shown as a proportion of tail DNA. Schematics outline primer-binding sites on the *Igk* locus (bottom of **a**, **b**). Error bars represent the standard deviation ($n = 4$ biologically independent samples for tail and FL pro-B cells $n = 5$ biologically independent samples for BM pre-B (CD19+B220+IgM−CD117+CD25+) and BM pro-B). P-values were calculated using two-tailed *T* tests. **c** 3-D immuno-DNA FISH was performed on ex vivo sorted B cells using BAC probes specific to the distal Vκ24 (RP23–101G13) gene region and the Cκ region (RP24–387E13), shown in red and green, respectively, in conjunction with an antibody to the phosphorylated form of γ-H2AX in white. Representative images of a B cell with no γ-H2AX associated with *Igk* alleles (top) and with γ-H2AX associated with one *Igk* allele (bottom). Scale bar = 1 μm. **d** Percentage of B cells (BM pro-B, BM pre-B, and FL pro-B) with at least one *Igk* allele associated with a γ-H2AX focus. Error bars represent standard deviation ($n = 2$ biologically independent samples). P-values were calculated using two-tailed Fisher's exact tests. **e** STAT5-ChIP-qPCR of iEκ on ex vivo sorted cells. The data are represented as a proportion of input DNA, and error bars represent the standard deviation ($n = 3$ biologically independent samples). For all P-values, *significant (0.05–0.01), **very significant (0.01–0.001), ***highly significant (<0.001). For **b**, **d**, and **e**, source data are provided as Source Data File

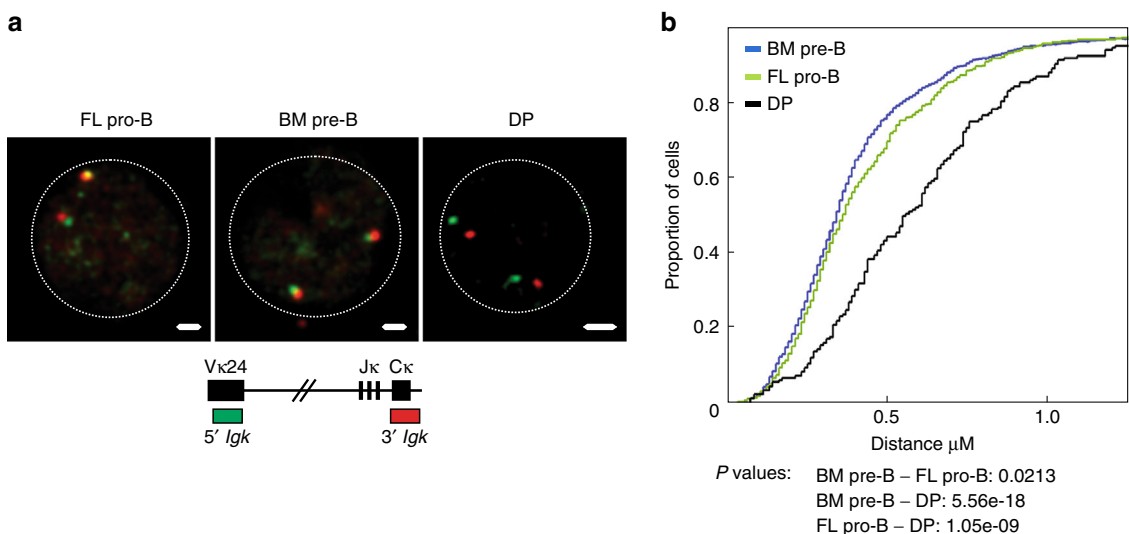

**Fig. 2** The *Igk* locus is similarly contracted in FL pro-B cells and BM pre-B cells. **a** Representative confocal microscopy images showing the distance separation between the probes on *Igk* in fetal liver pro-B cells (Lin−CD19+B220+IgM−CD2−CD117+), bone marrow pre-B cells (CD19+B220+IgM−CD117+CD25+), and double-positive T cells (CD90.2+TCRB−CD4+CD8+). **b** Distances separating the two ends of the locus are displayed as a cumulative frequency curve. A left shift on the curve is indicative of closer association. Bone marrow pre-B cells (blue), fetal liver pro-B cells (green), and negative control double-positive T cells (black). P-values were generated using two-sample Kolmogorov–Smirnov tests ($n = 2$ independent experiments)

Two different subsets of B-1a cells that segregate different functions of the B-1a cell compartment have been identified by the plasma cell alloantigen 1 (PC1)[59,60]. PC1lo B-1a cells produce higher amounts of natural IgM, and can undergo plasma cell differentiation, while PChi B-1a cells produce less natural IgM, have PtC-reactive BCRs, and can produce IL-10. In order to determine whether or not both of these subsets of B-1a cells were represented in B-1a cells generated in SLC-deficient mice, we analyzed PC1 expression. B-1a cells from *Igll1*−/− mice were found to have similar proportions of PC1lo and PC1hi B-1a cells as in wild-type mice (Fig. 4c). This suggests that not only is the B-1a compartment size normal in SLC knockout mice but also the PC1lo and PC1hi subsets are maintained.

B1-a cells cannot be detected in FL, and first appear in the transitional B-cell population 1 week after birth[61,62]. To further analyze the impact of surrogate light-chain deficiency on the generation of B1a cells, we examined transitional B-1a cells in spleens from 10–11 days old neonatal mice. As expected, the proportion of B-2 in cells in this population was severely reduced, while the proportion of B-1a cells was found to increase. This is reflected by the finding that total numbers of transitional B-2 cells are severely depleted compared with B1a cells in *Igll1*−/− versus

wild-type mice (Fig. 5a–c). Thus, even at this early stage of life, transitional B-1a cells from SLC-deficient mice are found at only moderately reduced levels compared with their wild-type counterparts, indicating that the presence of adult *Igll1*−/− B-1a cells is not a result of clonal expansion by self-renewal over time.

**B-1a cell development is inhibited in *Stat5b-CA* mice**. Our data suggest that low phospho-STAT5 signaling induces early *Igk* recombination in FL pro-B cells, which promotes the efficient generation of B-1a cells in SLC-deficient mice. To further validate the role of IL-7R/STAT5 signaling in B-1a cell development, we next asked whether increased STAT5 signaling could have the opposite effect on B-1a cell development and impair the generation of this compartment. To address this question, we used mice expressing a constitutive active form of STAT5 (*Stat5b-CA*)[63].

In these mice, we found that although B-2 and B-1b cell development were not significantly affected, all three independent littermate controlled *Stat5b-CA* mice exhibited a decrease in the B-1a cell compartment. This is reflected in cell proportions as well as the total cell numbers (Fig. 6a, b). Thus, STAT5 signaling inhibits B-1a cell development underscoring the importance of

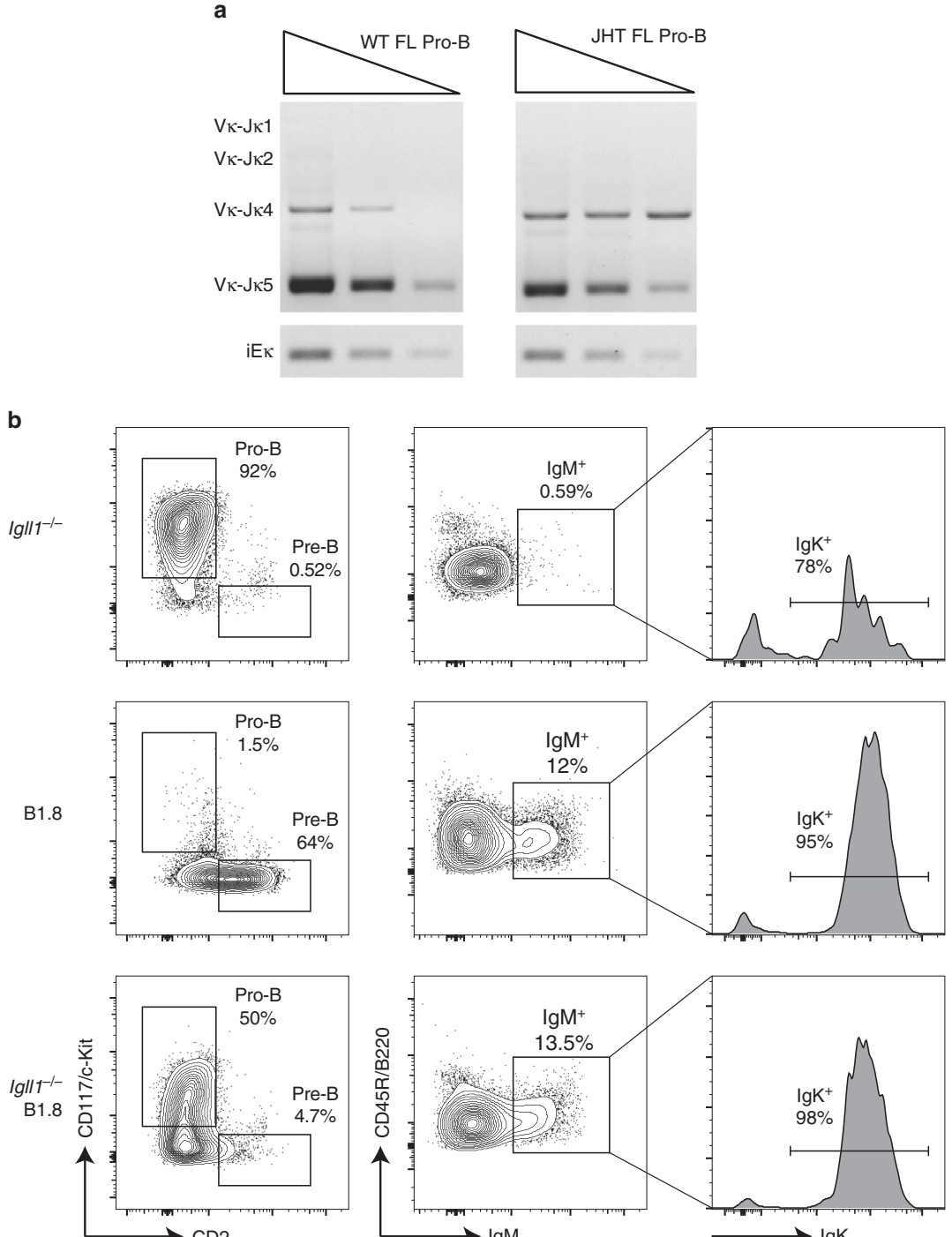

**Fig. 3** Early *Igk* rearrangement enables B cells to bypass the pre-BCR checkpoint. **a** *Igk* recombination can occur independent of *Igh* recombination as shown by semiquantitative PCR performed on ex vivo-derived fetal liver pro-B cells E17.5 (Lin⁻CD19⁺B220⁺IgM⁻CD2⁻CD117⁺) from wild-type and JHT mice. Each lane represents threefold serial dilutions of input DNA. DNA levels were normalized to iEκ **b** Representative flow-cytometry plots of fetal liver E17.5 cells from *Igll1*⁻/⁻, B1.8, and *Igll1*⁻/⁻;B1.8 littermates (from top to the bottom). Pro-B and pre-B cell gates are displayed as a percentage of CD19⁺IgM⁻ cells (left), IgM⁺ are displayed as a percentage of CD19⁺ cells (middle), and IgK⁺ histograms are displayed as a percentage of CD19⁺IgM⁺ cells (right)

reduced IL-7R/STAT5 signaling in driving the development of these cells.

**B-1a cells *Igll1*⁻/⁻ mice express PtC-specific receptors.** B-1a cells are known to have a strong bias for immunoglobulin rearrangements that confer specificity to PtC ($V_H12/V_K4$). In addition, the pre-BCR is known to select against B-1a-specific

autoreactive heavy chains as a result of a defect in pairing of the latter with SLC[26,27]. Given these facts, we next asked whether B-1a cells that develop independent of SLC would promote the generation of B-1a-specific rearrangements. To address this question, we analyzed peritoneal cavity B cells for $V_H12$ usage and specificity against PtC. As shown in Fig. 6c, B-1a cells from *Igll1*⁻/⁻ mice express $V_H12$ rearranged receptors. Importantly, these $V_H12^+$ B-1a cells bind PtC containing liposomes, inferring

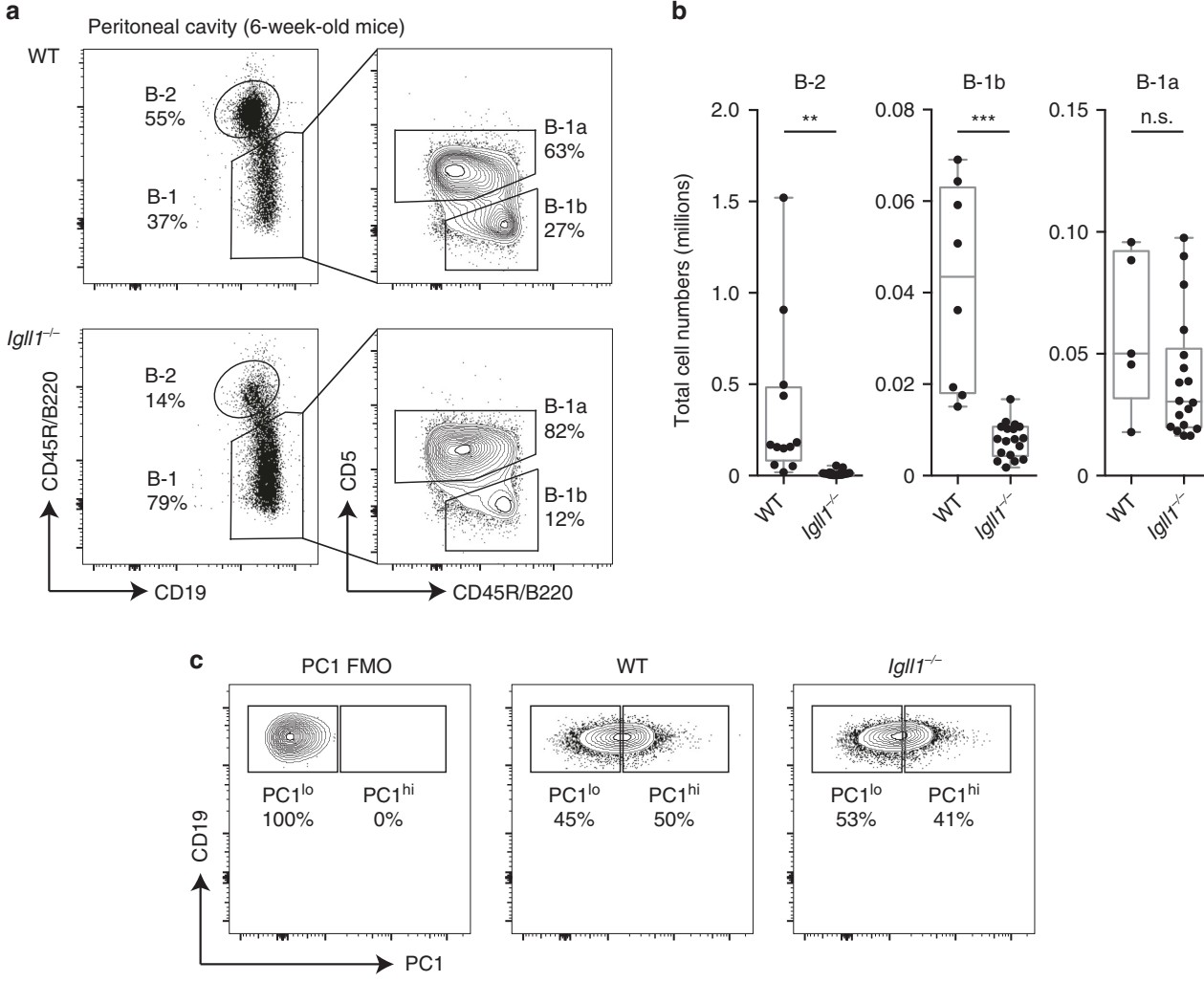

**Fig. 4** B-1a cells are efficiently generated in surrogate light chain-deficient mice. **a** Representative flow-cytometry plots of peritoneal cavity cells from wild-type and $Igll1^{-/-}$ mice. B-2 and B-1 cell gates are displayed as a percentage of CD19$^+$ cells (left). B-1a and B-1b cell gates are displayed as a percentage of B-1 cells (right). **b** The total cell numbers of B-2, B-1b, and B-1a cells were calculated from the peritoneal cavity of these mice. The middle line in the boxplots represents the median. The box shows first and third quartiles, and the whiskers represent the max and min values. Each dot represents cells from a different individual mouse. *P*-values were calculated using a two-tailed *T* test. Source data are provided as Source Data File. **c** $Igll1^{-/-}$ mice maintain both functionally distinct PC1$^{lo}$ and PC1$^{hi}$ subsets of B-1a cells. Representative flow-cytometry plots of peritoneal cavity cells from wild-type and $Igll1^{-/-}$ mice. B-2 and B-1 cell gates are displayed as a percentage of CD19$^+$ cells (left). PC1$^{lo}$ and PC1$^{hi}$ B-1a cells are displayed as a percentage of B-1a cells (right). A PC1 FMO (fluorescence minus one) was used to define the gate

that they have the canonical $V_H12/V_K4$ gene segment bias (Fig. 6c). In addition, there is not a significant difference in the proportion of B-1a cells that have specificity against PtC (Fig. 6d). Thus, an absence of SLC supports the generation of these self-reactive B-1a cells.

**$Igll1^{-/-}$ and wild-type B-1a cells have similar repertoires**. To determine how SLC deficiency affects gene segment usage, we sequenced the BCR repertoires of B-1a cells from WT and SLC-deficient mice. Briefly, peritoneal cavity B-1a cells were sorted, lysed, and the rearranged *Igh* products amplified by PCR. Illumina adapters and barcodes were added, and the rearrangements sequenced by next-generation sequencing. BCR repertoires were analyzed by IMGT/High-V-QUEST, and visualized using IMGT/StatClonotype[64–67]. These analyses revealed that B-1a cells from $Igll1^{-/-}$ mice have a similar overall $V_H$ usage compared with those of wild-type mice (Fig. 7). This shows that B-1a cells that develop in the absence of SLC have a wide distribution of gene

segment usage, and thus do not represent a clonal expansion of a few rare progenitors. A diverse repertoire was also observed for *light-chain* gene segment usage (Fig. 8). As expected, $V_H12$ ($V_H12$-3) usage was found to be high in wild-type B-1a cells, and calculated proportions were comparable with our flow-cytometry experiments in the cells from $Igll1^{-/-}$ mice (Fig. 6c)[7]. $V_K4$, which pairs with $V_H12$ to confer PtC specificity, has a reduced representation in SLC-deficient B-1a cells, but it is still found at significantly higher frequency than in wild-type B-2 cells (Fig. 8). These rearrangement studies support the data from our flow-cytometry analysis and underscore the finding that B-1a cells generated in the absence of SLC maintain a $V_H12/V_K4$ gene segment bias.

To confirm that B-1a cells generated in SLC-deficient mice were from fetal origin, we analyzed *Igh* sequence for N-additions. B-1a cells are known to predominantly come from fetal origins and as a consequence of low Tdt activity at this stage, they are known to have reduced N-additions compared with adult bone marrow-derived B cells. Our analysis revealed that $Igll1^{-/-}$

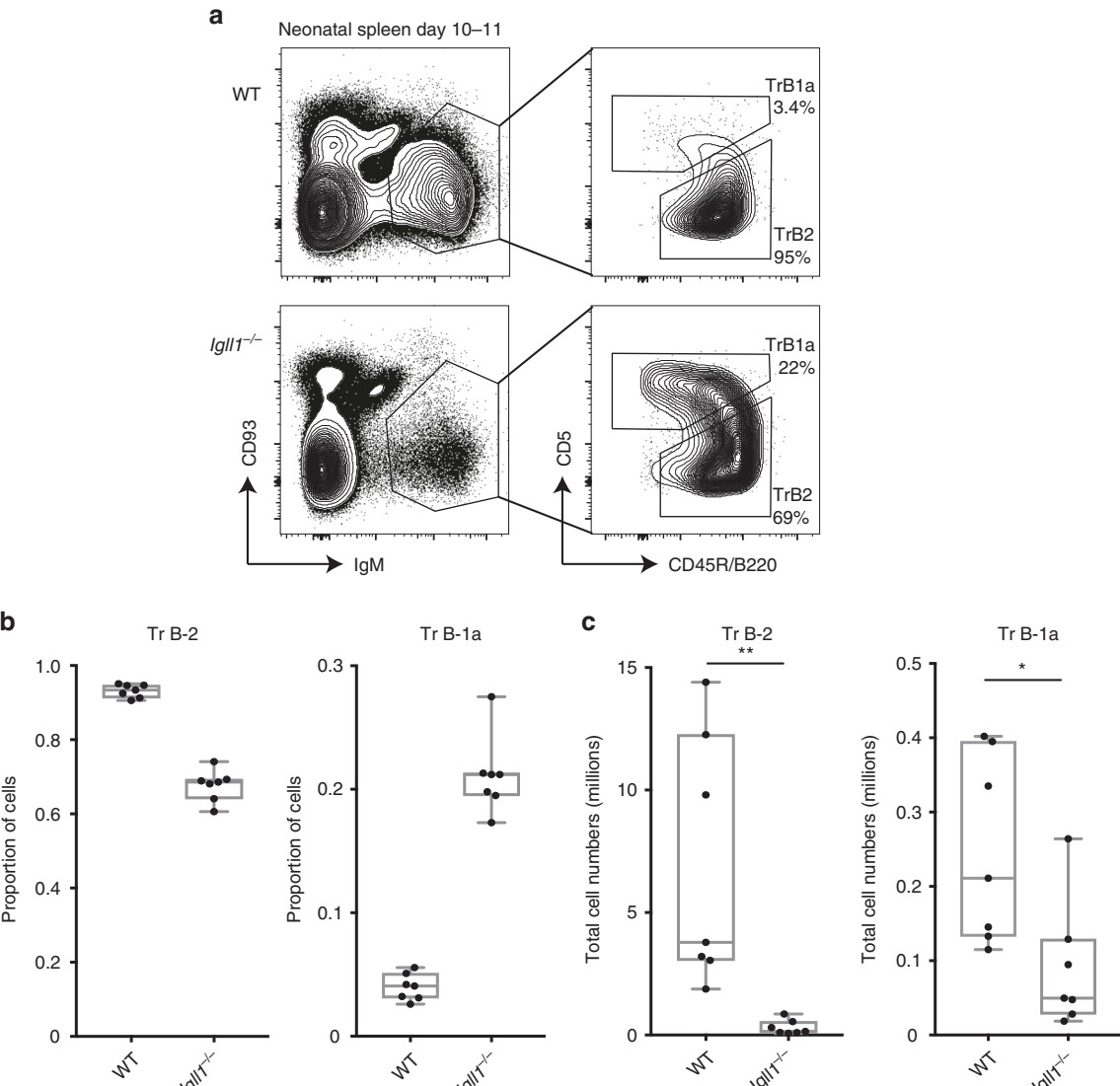

**Fig. 5** Transitional B-1a cells are readily detected in neonatal spleens from *Igll1−/−* mice. **a** Representative flow-cytometry plots showing transitional B-1a (CD19+CD93lo/intIgM+B220−/loCD5+) and transitional B-2 (CD19+CD93lo/intIgM+CD5−/loB220+) from wild-type and *Igll1−/−*. The left plot is gated on CD19+ cells from D10–11 neonatal spleen. **b** Graphs represent the proportion of cells that are either transitional B-1a or transitional B-2. **c** Graphs show the total cell numbers of transitional B-2 or transitional B-1a in millions. For both **b** and **c**, the middle line in the boxplots represents the median. The box shows first and third quartiles, and the whiskers represent the max and min values. Each dot represents cells from a different individual mouse. Source data are provided as Source Data File

peritoneal cavity B-1a cells had similarly reduced levels of N-additions as wild-type controls (Fig. 9). The paucity of N-additions supports the idea that B-1a cells are efficiently generated in SLC-deficient mice, maintaining key B-1a cell traits like $V_H12$ and $V_K4$ gene segment bias and reduced levels of junctional diversity.

## Discussion

Although it was known that B-1a cells predominantly develop in the FL, the pathways driving development and the mechanisms underlying the generation of B cells with a repertoire skewed toward autoreactivity were not previously known. Indeed, this has been a subject of debate for many years. Here, we now reveal that a reduction in IL-7 signaling in the FL environment alleviates *Igk* repression and promotes early rearrangement of this locus in pro-B cells. Productive *heavy* and *light-chain* gene rearrangement at the pro-B cell stage can lead directly to the expression of a mature

BCR, thereby bypassing the requirement for a pre-BCR check-point. Since PtC-specific $V_H$ gene rearrangements pair poorly with SLC[26] and the pre-BCR selects against autoreactive receptors[27], SLC-independent development provides an explanation for how B cells with autoreactive, PtC-specific receptors are generated. In more general terms and along Jerne's idea that the immune system selects antibody mutants from an initial self-reactive repertoire[68], B-cell progenitors can be initially positively selected by the expression of autoreactive BCRs or their evolutionary surrogate, the pre-BCR[32,33]. This is followed by negative selection of autoreactivity through receptor editing. The skewing of the B-1a receptor repertoire toward autoreactivity likely reflects the imprint of the initial positive selection of B-1a cells through autoreactive BCRs rather than SLC (Fig. 10).

The idea that B-1 cells can develop in a SLC-independent manner is not novel. In fact, Kitamura et al. observed that in contrast to B-2 cells, B-1 cells were essentially unaffected by SLC deficiency, however, the distinct effect on the B-1a versus B-1b

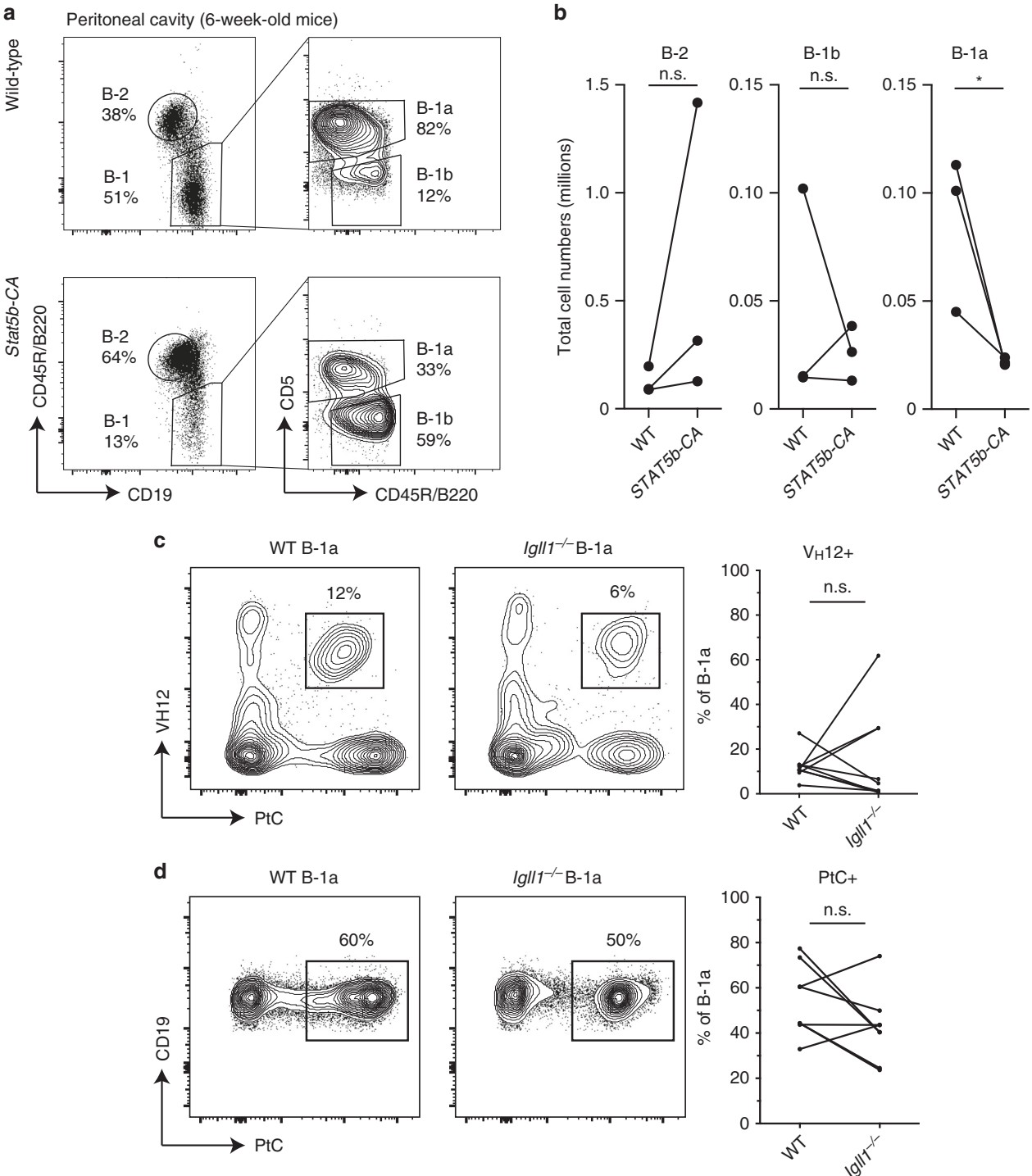

**Fig. 6** Constitutive phospho-STAT5 signaling selectively inhibits B-1a cell development. **a** Representative flow-cytometry plots of peritoneal cavity cells from wild-type and *Stat5b-CA* mice. B-2 (CD19+B220hi) and B-1 (CD19+B220lo) cell gates are displayed as a percentage of CD19+ cells (left). B-1a (CD19+B220loCD5+) and B-1b (CD19+B220loCD5−) cell gates are displayed as a percentage of the B-1 cells (right). **b** Graphs display the total cell numbers of B-2, B-1b, and B-1a cells from the peritoneal cavity of wild-type and *Stat5b-CA* mice. Each dot represents an individual mouse, and lines connect pairs of littermates from three different litters of mice. **c, d** SLC-independent B-1a cells express receptors with phosphatidylcholine (PtC)-specific V_H12 gene rearrangements. Peritoneal cavity B-1a cells from wild-type and *Igll1−/−* mice highlight the proportion of B-1a cells that are either **c** V_H12+, PtC+ or **d** PtC+. The left side shows representative flow-cytometry plots. The right side is a graphical summary of the mice from all the experiments. Each dot represents an individual mouse and lines connect pairs of littermates. *P*-values were calculated using a two-tailed *T*-test. Source data are provided as Source Data File

cell compartment was not investigated[28]. Our analyses demonstrate that while B-2 and B-1b cell development is severely impaired in *Igll1−/−* mice, B-1a cells can be generated at wild-type levels. Furthermore, SLC-independent B-1a development can

occur through a mechanism that involves reduced IL-7R/STAT5 signaling and early *Igk* rearrangement. We found that B-1a cell development is selectively impaired in *Stat5b-CA* mice, providing additional support for the idea that SLC-independent

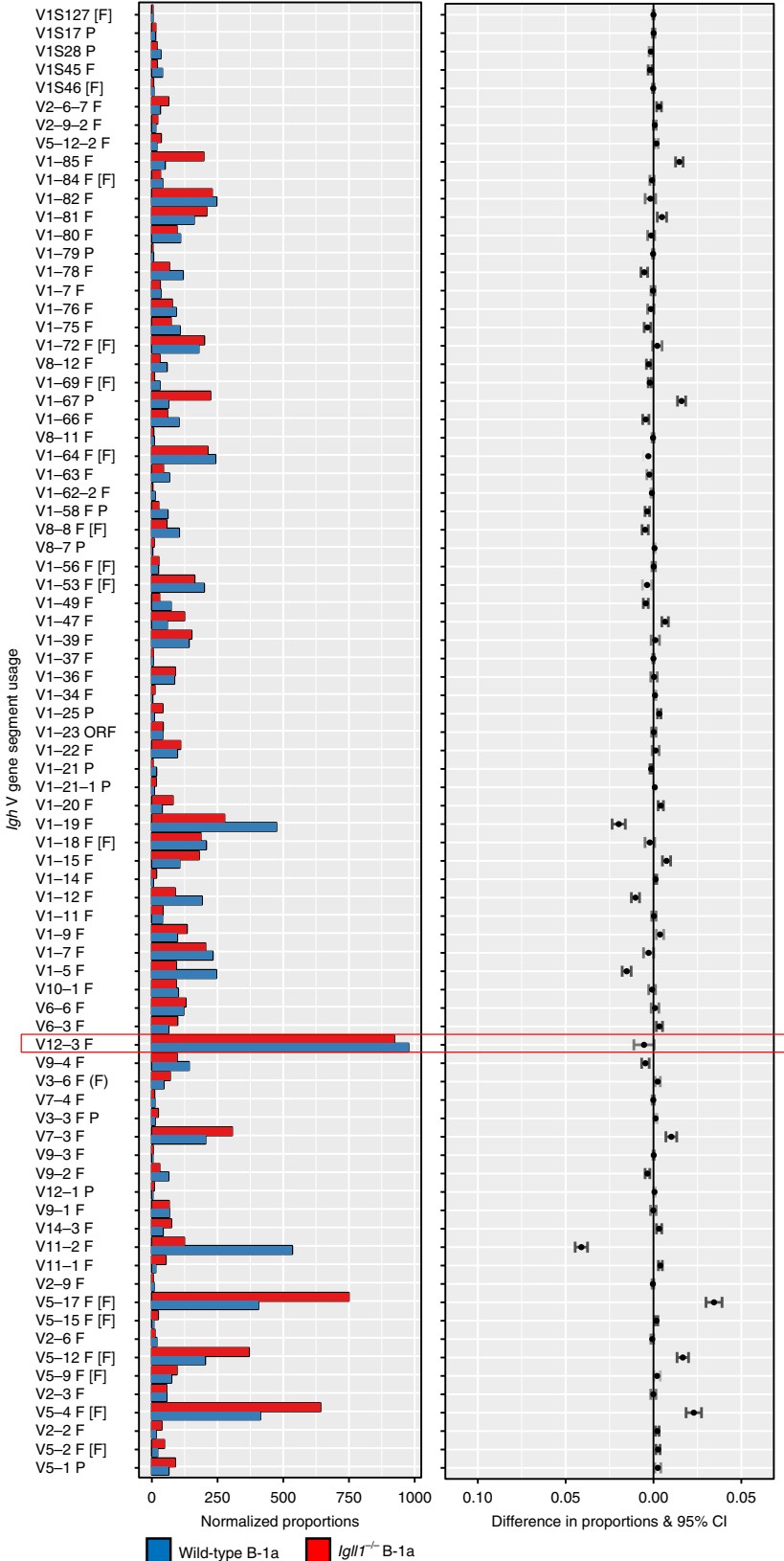

**Fig. 7** $Igll1^{-/-}$ B-1a cells have a similar $V_H$ gene segment $V_H$ usage to wild-type controls. Representative synthesis graphs generated from IMGT/StatClonotype that compare differences in V gene segment usage between wild-type (blue) and $Igll1^{-/-}$ (red)-derived B-1a cells (CD19$^+$B220$^{lo}$CD5$^+$). The red box highlights $V_H12$ ($V_H12$-3) gene segment ($n = 2$ independent experiments). The graph combines a bar graph for the normalized proportions of each gene segment and the differences in proportions with significance and confidence intervals (CI)

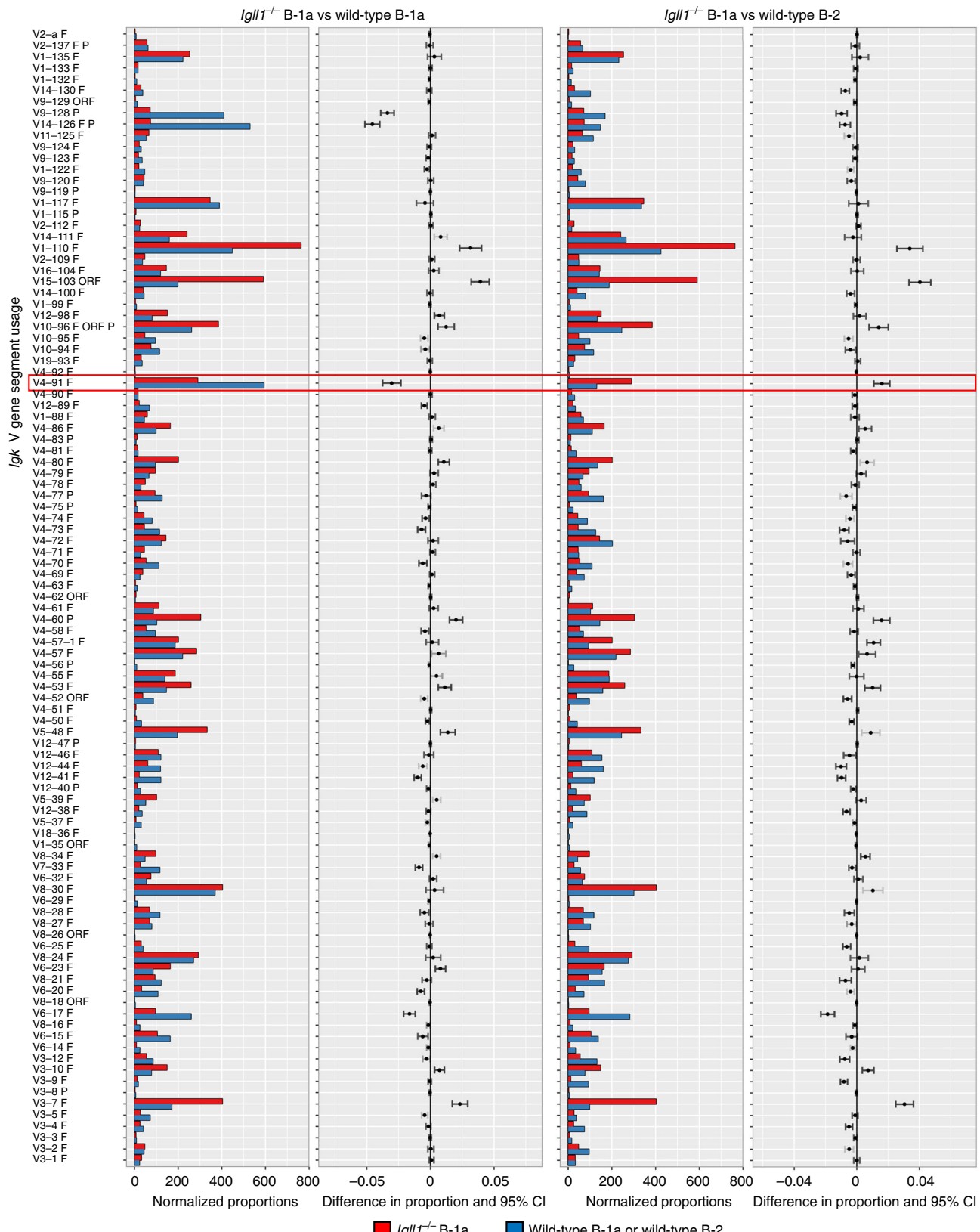

**Fig. 8** Wild-type and *Igll1*⁻/⁻ *Igll1*⁻/⁻ B-1a cells have a similar V_K gene segment usage to wild-type controls and maintain V_K4-91 enrichment. Representative synthesis graphs generated from IMGT/StatClonotype that compare differences in *Igk* V gene segment usage between *Igll1*⁻/⁻ (red) versus wild-type (blue) peritoneal cavity B-1a cells CD19⁺B220^loCD5⁺ (left side) or *Igll1*⁻/⁻ (red) peritoneal cavity B-1a cells vs wild-type (blue) peritoneal cavity B-2 cells CD19⁺B220⁺ (right side). The red box highlights V_K4 (V_K4-91) gene segment (*n* = 2 independent experiments). This graph combines a bar graph for the normalized proportions of each gene segment and the differences in proportions with significance and confidence intervals (CI)

B-1a cell development is favored by a decrease in IL-7 signaling. It should be noted that although *Igk* rearrangement can occur prior to *Igh* rearrangement, it is not a prerequisite for SLC-independent B-1a cell development. Indeed, a rearranged *Igh* that cannot associate with SLC could persist at the pro-B cell stage until subsequent productive *Igk* rearrangement generates a BCR independent of IGH-SLC pairing. Taken together, our findings point to a model in which B-1a cells typically develop in a SLC-independent manner. Interestingly, earlier work analyzing expression patterns of genes relevant to rearrangement in the yolk

sac, para-aortic splanchnopleura, and spleen demonstrates that at E9–11, early lymphoid progenitor cells express *Rag2* and *VpreB*, but lack expression of *Igll1*[69]. The delayed expression of the SLC component, lambda5 (*Igll1*), points to the possibility that there is an early window in which B-cell development can occur in the absence of SLC. This stage coincides with the stage at which the earliest B-1 progenitor cells are detected (E9 in the yolk sac and para-aortic splanchnopleura)[70,71]. Thus, we speculate that the reason why early progenitors become B-1 cells is because they develop in a SLC-independent manner. The observation that SLC-deficient mice generate B-1a cells with autoreactive receptors supports this model[27]. The generation of SLC-independent autoreactive receptors does not need to occur at high frequency, because cells that acquire a B1a cell receptor undergo a rapid proliferative burst and thus even a few cells have the ability to give rise to a robust B1a cell population[20]. In sum, assignment of a SLC-independent pathway of B-cell development to B-1a cells provides a new perspective on B-1a development, reconciling old lineage and selection models (Fig. 10).

## Methods

**Mice**. *Igll1*[−/−] (Jackson Laboratory 002401) and *Stat5b-CA* mice were maintained on a C57BL/6 background, as previously described[28,36,72]. B1.8 mice[73] were crossed onto a *Igll1*[−/−] background. These mice were housed and cared for in accordance with IACUC guidelines and protocols approved by NYUMC (protocol #: IA15-01468) and U of Minnesota (protocol #: 1502–32347A).

**Flow cytometry and antibodies**. BM and FL cell populations were isolated from C57Bl/6 mice via cell sorting and analyzed by flow cytometry. B- and T-cell antibodies include: anti-CD45R/B220 (clone: RA3–6B2 cat. no. 553092), anti-CD19 (clone 1D3 cat. no. 557655), anti-IgM[b] (clone AF6–78 cat. no. 553520), anti-IgM (clone II/41 cat. no. 553437, 743324), anti-IgK (clone 187.1 cat. no. 562476, 562888), anti-CD117/c-Kit (clone 2B8 cat. no. 553356), anti-CD2 (clone RM2–5 cat. no. 553111), anti-CD25/IL2RA (clone PC61 cat. no. 553866), anti- CD93 (clone AA4.1 cat. no. 561990) anti-CD90.2/Thy1.2 (clone 53–2.1 cat. no. 25-0902-82), anti-CD5 (clone 53-7.3 cat. no. 553022), anti-V$_H$12 (clone 5C5)[74] anti-TCRB (clone H57–597 cat. no. 47-5961-82), anti-CD8a (clone 53-6.7 cat. no. 553031), anti-CD4 (clone RM4-5 cat. no. 553051), anti-ENPP1/PC1 (clone YE1/19.1 cat. no. 149207). All of

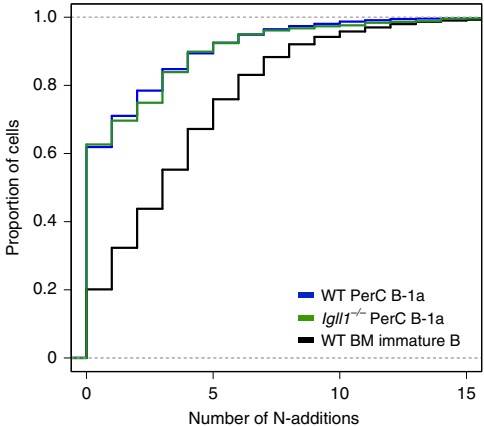

**Fig. 9** Wild-type and *Igll1*[−/−] peritoneal cavity B-1a cells have similar low levels of N-additions as compared with wild-type counterparts. Cumulative frequency graph demonstrates the proportion of cells that have a certain number of N-additions between V$_H$ and V$_D$ gene segments for wild-type peritoneal cavity B-1a cells CD19+B220$^{lo}$CD5+ (blue), *Igll1*[−/−] peritoneal cavity B-1a CD19+B220$^{lo}$CD5+ cells (green), and wild-type bone marrow immature B cells CD19 + B220 + IgM$^{int}$ (black) (n = 2 independent experiments)

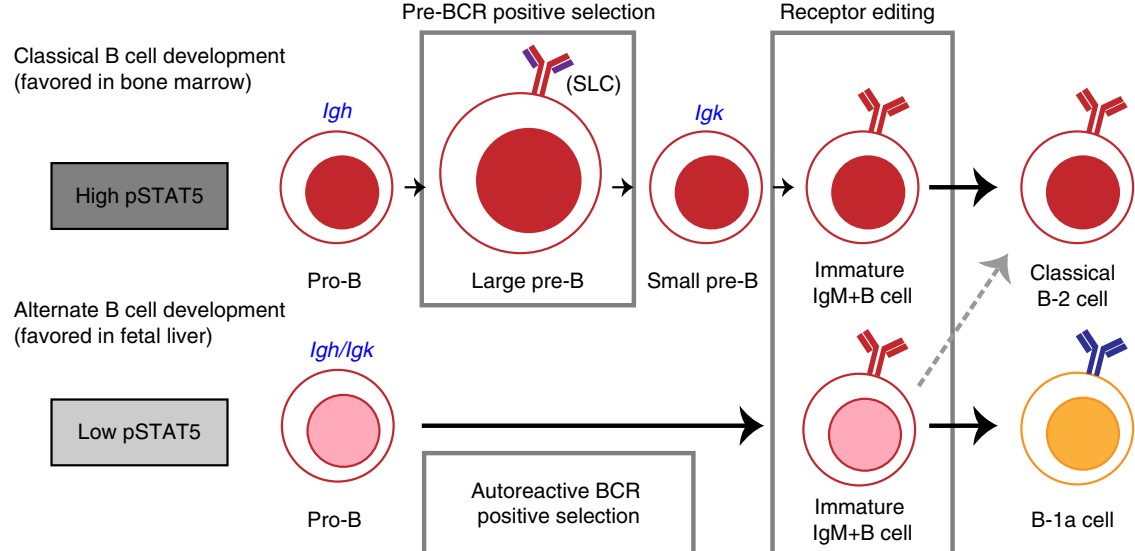

**Fig. 10** The model of classical versus alternate B-cell development. The classical B-cell developmental pathway is favored in the bone marrow where high levels of STAT5 signaling inhibit *Igk* recombination at the pro-B cell stage. At the large pre-B cell stage, a productive heavy-chain pairs with SLC and pre-BCR signaling at this stage leads to proliferative expansion and positive selection. Autoreactive heavy-chains pair poorly with SLC, and will not experience positive selection in this manner. Following *Igk* recombination (blue) at the small pre-B cell stage, self-reactive B cells can undergo receptor editing to further select against self-reactive BCRs. The alternate B-cell developmental pathway is favored in fetal liver where low levels of STAT5 signaling in pro-B cells promotes *Igk* recombination at the same stage as *Igh* recombination (blue). This enables the generation of B cells expressing a mature BCR instead of a pre-BCR. Here, self-reactive B cells are initially positively selected by self-antigens. Although these B cells can potentially undergo receptor editing to modify their BCRs, the initial positive selection of autoreactive B cells skews the B-1a BCR repertoire toward autoreactivity

the antibodies were obtained from BD Pharmigen, except anti-CD90.2, anti-TCRB, and anti-ENPP1/PC1 (eBioscience). Fluorescent DPOC/CHOL liposomes were acquired from FormuMax. The gating strategy was as follows: bone marrow pro-B cells B220$^+$/CD19$^+$/IgM$^-$/c-kit$^+$/CD25$^-$/IgM$^-$, BM pre-B cells B220$^+$/CD19$^+$/IgM$^-$/c-kit$^-$/CD25$^+$, FL pro-B cells (E17.5) B220$^+$/CD19$^+$/IgM$^-$/c-kit$^+$/CD2$^-$, and thymic DP cells Thy1.2$^+$/TCRB$^-$/CD19$^-$/CD4$^+$/CD8$^+$. FL pro-B cells were sorted on CD2$^-$ instead of CD25$^-$, because CD25 is not expressed in fetal B cells. CD2 expression is correlated with cytoplasmic μ heavy chain, and can be used to identify pre-B cells and efficiently sort pro-B cells from the FL[75]. From the peritoneal cavity, B-2, B-1a, and B-1b were identified as followed: B-2 cells B220$^{hi}$/CD19$^+$, B-1a cells B220$^{lo}$/CD19$^+$/CD5$^+$, and B-1b cells B220$^{lo}$/CD19$^+$/CD5$^-$. To obtain FL precursor cells, FLs from E15.5–17.5 were subject to lineage depletion using the following biotin-labeled antibodies: anti-CD4 (RM4–5), anti-CD8a (53-6.7), anti-TCRγδ, anti-CD49b (DX5), anti-Ly6G/Ly6C (RB6-8C5), anti-CD11c (N418), anti-CD11b (M1/70), anti-F4/80 (BM8), anti-Ter-119 (TER-119) followed by a negative selection using streptavidin magnetic rapidspheres (STEMCELL Technologies). The remaining cells were sorted as: Lin$^-$/CD19$^-$/IgM$^-$/c-kit$^+$/CD2$^-$/B220$^{lo}$. Cells were sorted using a FACSAria I (BD). The data were also collected on an LSR II (BD) and analyzed using FlowJo software.

**Intracellular staining for IgK**. FL E17.5 B cells were stained for surface markers (CD19, B220, IgM, IgK, CD2, and CD127), followed by permeabilization/fixation and secondary staining for IgM and IgK in accordance with the manufacturer's instructions (BD 554714). Antibodies with different conjugated flourophores were used to distinguish between surface IgM/IgK (BD 554714 and 562476) and intracellular IgM/IgK (BD 553437 and 562888).

**Semiquantitative *Igk* rearrangement**. DNA was prepared from sorted cells by proteinase K digestion at 55 C for 4 h, 85 C for 10 min. VκJκ joints were amplified using a degenerate Vκ primer (5′-GGCTGCAGSTTCAGTGGCAGTGGRTCWG-GRAC-3′), and a Jκ5 primer (5′-ATGCCACGTCAACTGATAATGAGCCCTC TCC-3′). *Acidia* gene was amplified as a loading control (AID F: 5′-GCCACCTT CGCAACAAGTCT-3′, AID R: 5′-CCGGGCACAGTCATAGCAC-3′). Bands were run through a 1.5% agarose gel electrophoresis and imaged using a ChemiDoc XR + (Bio-Rad). Germline retention analysis of Jκ1 was carried out by real-time quantitative PCR analysis, as previously described[50].

**Immuno-DNA FISH**. Combined detection of γ-H2AX and *Igk* loci was carried out on cells adhered to poly-L lysine-coated coverslips, as previously described[51]. Cells were fixed with 2% paraformaldehyde/PBS for 10 min, and permeabilized for 5 min with 0.4% Triton/PBS on ice. After 30 min blocking in 2.5% BSA, 10% normal goat serum, and 0.1% Tween-20/PBS, H3S10ph staining was carried out using an antibody against phosphorylated serine-10 of H3 (Millipore) diluted at 1:400 in blocking solution for 1 h at room temperature. Cells were rinsed three times in 0.2% BSA, 0.1% Tween-20/PBS and incubated for 1 h with goat-anti-rabbit IgG Alexa 488 or 594 or 633 (Invitrogen). After three rinses in 0.1% Tween-20/PBS, cells were post fixed in 3% paraformaldehyde/PBS for 10 min, permeabilised in 0.7% Triton-X-100 in 0.1 M HCl for 15 min on ice, and incubated in 0.1 mg/ml RNase A for 30 min at 37 °C. Cells were then denatured with 1.9 M HCl for 30 min at room temperature and rinsed with cold PBS. DNA probes were denatured for 5 min at 95 °C, pre-annealed for 45 min at 37 °C, and applied to coverslips which were sealed onto slides with rubber cement and incubated overnight at 37 °C. Cells were then rinsed three times 30 min with 2x SSC at 37 °C, 2x SSC, and 1x SSC at room temperature. Cells were mounted in ProLong Gold (Invitrogen) containing DAPI to counterstain the total DNA.The *Igk* locus was detected using BAC DNA probes that hybridize to the distal Vκ24 gene region (RP23–101G13) and the Cκ region (RP24–387E13), in combination with an antibody against the phosphorylated form of γ-H2AX. BAC probes were directly labeled by nick translation with dUTP-A594 or dUTP-A488 (Invitrogen). FISH for locus contraction was conducted as previously published[51].

**Confocal microscopy and analysis**. Cells were analyzed by confocal microscopy on a Leica SP5 Acousto-Optical Beam Splitter system. Optical sections separated by 0.3 μm were collected. Analysis on cells included those in which signals from both alleles could be detected, which encompassed 90–95% of the total cells imaged. Further analysis was carried out using ImageJ. Alleles were defined as colocalized with γ-H2AX if the signals overlapped. Sample sizes typically included a minimum of 100 cells, and experiments were repeated at least two or three times. Statistical significance was calculated by $\chi^2$ analysis in a pair-wise manner. For locus contraction, distances were measured between the center of mass of each BAC signal. Significant differences in distributions of empirical inter-allelic distances were determined by a nonparametric two-sample Kolmogorov–Smirnov (KS) test. To eliminate observer bias, each experiment was analyzed by at least two people.

**STAT5-ChIP-qPCR**. STAT5-ChIP-qPCR was conducted as previously described[44]. Purified DNA was then analyzed by quantitative real-time PCR. Samples were analyzed in triplicate and represent three biological replicates.

**Igh and Igk repertoire analysis**. Cells were sorted and lysed in 0.5 mg/mL proteinase K at 55 C for 4 h, 85 C 10 min. Lysed cells were used as a template for HotStarTaq (Qiagen) PCR amplification. *Igh* primers used to amplify rearrangements were adapted from V$_H$ FR3 and J$_H$ BIOMED-2 primers[76,77]. Additional primers were designed to capture some V genes missing from the BIOMED-2 study, which include V$_H$12 (V$_H$12-3). Igk primers include the degenerate V primer used in the semiquantitative PCRs with additional primers designed to capture some B-1a cell-biased gene segments, which include V$_K$4 (V$_K$4–91) (Supplementary Table 2). Amplified rearrangements were purified by gel extraction (Qiagen) followed by a standard end-repair reaction. Following purification steps were carried out by Ampure XP bead purification (Beckman Coulter). In order to attach Illumina-compatible adapters, samples were treated by standard dA tailing followed by adapter ligation using Quick Ligase (NEB) mixed with pre-annealed NEXTflex DNA Barcodes (Bioo Scientific). QC was carried out by tapestation, and quantified by qPCR (Kapa Biosystems). Samples were pooled and sequenced on an Illumina HiSeq 4000 (2 × 150 PE reads). Nucleotide sequences were compared with the reference equences from IMGT, the international ImMunoGeneTics information system (http://www.imgt.org) and analyzed using IMGT/HighV-QUEST[67], a web portal allowing the analysis of thousands of sequences on IMGT/V-QUEST[65]. IMGT/StatClonotype was used to analyze statistically significant gene segment usage between samples[64,66]. *Igh* N-addition information from the region between V and D gene segments was extracted from IMGT/HighV-QUEST and graphed in R.

**Reporting summary**. Further information on research design is available in the Nature Research Reporting Summary linked to this article.

## Data availability

The data that support findings of this study have been deposited in the Gene Expression Omnibus (GEO) database under the accession code GSE104111.

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

## Acknowledgements

The authors would like to thank the NYU Flow Cytometry and Cell Sorting Center as well as the NYU Genome Technology Center, both of which are supported in part by the Cancer Center Support Grant P30CA016087 at the Laura and Isaac Perlmutter Cancer Center. We would like to thank Meinrad Busslinger and Taras Kreslavsky (Research Institute of Molecular Pathology (IMP)) for suggesting the transitional B-1a cell experiment. In addition, we would like to thank the Skok lab for discussions on the study. J.A.S. is supported by NIH grants R35GM122515. J.B.W. was previously supported by the T32 CA009161 training grant (Levy) and the 2T32 AI100853-06 (Reizis) training grant. S.L.H. was supported by an American Society of Hematology (ASH) Scholar Award and by a Molecular Oncology and Immunology Training Grant NIH T32. M.A.F. is supported by NIH grants CA151845 and CA154998.

## Author contributions

J.B.W., S.L.H., K.J., and J.A.S. designed the experiments. J.B.W., S.L.H., L.M.H.-H., M.M., and K.J. performed the experiments and conducted data analysis. J.B.W., J.A.S., and K.R. wrote the paper, with input from M.M., S.B.K., M.R.C., and M.A.F.

## Competing interests

The authors declare no competing interests.
