## [Peer Review File · Nature Communications]

Editorial Note: This manuscript has been previously reviewed at another journal that is not operating a transparent peer review scheme. This document only contains reviewer comments and rebuttal letters for versions considered at Nature Communications .

REVIEWERS' COMMENTS:

Reviewer #1 (Remarks to the Author):

This Ms documents significant differences in how fetal liver-derived B1a B cells develop compared to bone marrow-derived B cells. The findings will be of interest to both B cell biologists and more generally to immunologists (as this serves as an important and interesting example of mechanisms by which the antigen-specific receptor participates in fate determination of lymphocyte lineages). The major findings of the paper are for the most part novel.

Major claims of the paper :

- 1) " in the fetal liver (FL) versus bone marrow (BM) environment, reduced IL-7R/STAT5 levels promote immunoglobulin kappa (Igk) gene recombination at the early pro-B cell stage.
- 2) [fetal-derived] pro-B cells can directly generate a mature B cell receptor (BCR) and bypass the requirement for a pre-BCR and pairing with surrogate light chain (SLC)."

The work is quite convincing, using mouse genetic models to support the findings as well as appropriate molecular tools. I believe that it will influence thinking in this field, as it tends to – to some extent -unify the prevalent models – a) that B1a cells are derived from a specific lineage-progenitor vs b) B1a cells are selected into that cell type via the specificity of their receptors. It also provides a plausible explanation for the self-reactivity of B1a cells – the fact that they can by-pass the pre-BCR check-point.

Reviewer #2 (Remarks to the Author):

The manuscript by Wong et al show evidence that offer a mechanism to explain why B1a cells with particular functional properties are produced in fetal and neonatal life.

It is shown the in normal fetal liver pro-B cells have higher levels of recombined IgK than in bone marrow presumably in the absence of Igh productive rearrangement. To show that these cells can bypass pre-BCR selection the authors analyze surrogate light chain (SLC) deficient mice where all B cell are produced through an early rearrangement of the light chain. That the few B cells produced in SLC deficient mice bypass pre-BCR selection due to an early light chain rearrangement has been previously shown. It has also previously been shown that light chain rearrangement is regulated by the availability of IL-7 (Johnson et al J. Immunol.) and the mechanisms leading to that also analyzed (Fistonish et al JEM 2018). It has also been shown that in the absence of SLC autoreactive B cells are not counter-selected because they bypass the pre-BCR selection (Keenan et al Science 2008)

The difficulty here is to show that this is what happens in normal animals in the generation of the B1a compartment due to low amounts of IL7 in fetal liver. To accumulate evidence for that the authors show that in SLC ko mice, although reduced, B1a cells are less so than B2 cells and that constitutive STAT5 expression the B1a compartment is more reduced than the B2 compartment. This is still indirect evidence and remains correlative. The ideal would be to show that, not in excess of STAT5 there is a reduction, but that in the absence of IL7 there is an increased production of transitional B1a cells shortly after birth similar to the experiment shown in SLC deficient mice.

Reviewer #4 (Remarks to the Author):

N/A

REVIEWERS' COMMENTS:

Reviewer #1 (Remarks to the Author):

This Ms documents significant differences in how fetal liver-derived B1a B cells develop compared to bone marrow-derived B cells. The findings will be of interest to both B cell biologists and more generally to immunologists (as this serves as an important and interesting example of mechanisms by which the antigen-specific receptor participates in fate determination of lymphocyte lineages). The major findings of the paper are for the most part novel.

Major claims of the paper :

- 1) “ in the fetal liver (FL) versus bone marrow (BM) environment, reduced IL-7R/STAT5 levels promote immunoglobulin kappa (Igk) gene recombination at the early pro-B cell stage.
- 2) [fetal-derived] pro-B cells can directly generate a mature B cell receptor (BCR) and bypass the requirement for a pre-BCR and pairing with surrogate light chain (SLC).”

The work is quite convincing, using mouse genetic models to support the findings as well as appropriate molecular tools. I believe that it will influence thinking in this field, as it tends to – to some extent -unify the prevalent models – a) that B1a cells are derived from a specific lineage-progenitor vs b) B1a cells are selected into that cell type via the specificity of their receptors. It also provides a plausible explanation for the self-reactivity of B1a cells – the fact that they can by-pass the pre-BCR check-point.

Reviewer #2 (Remarks to the Author):

The manuscript by Wong et al show evidence that offer a mechanism to explain why B1a cells with particular functional properties are produced in fetal and neonatal life.

It is shown the in normal fetal liver pro-B cells have higher levels of recombined IgK than in bone marrow presumably in the absence of Igh productive rearrangement. To show that these cells can bypass pre-BCR selection the authors analyze surrogate light chain (SLC) deficient mice where all B cell are produced through an early rearrangement of the light chain. That the few B cells produced in SLC deficient mice bypass pre-BCR selection due to an early light chain rearrangement has been previously shown. It has also previously been shown that light chain rearrangement is regulated by the availability of IL-7 (Johnson et al J. Immunol.) and the mechanisms leading to that also analyzed (Fistonish et al JEM 2018). It has also been shown that in the absence of SLC autoreactive B cells are not counter-selected because they bypass the pre-BCR selection (Keenan et al Science 2008)

The difficulty here is to show that this is what happens in normal animals in the generation of the B1a compartment due to low amounts of IL7 in fetal liver. To accumulate evidence for that the authors show that in SLC ko mice, although reduced, B1a cells are less so than B2 cells and that constitutive STAT5 expression the B1a compartment is more reduced than the B2 compartment. This is still indirect evidence and remains correlative. The ideal would be to show that, not in excess of STAT5 there is a reduction, but that in the absence of IL7 there is an increased production of transitional B1a cells shortly after birth similar to the experiment shown in SLC deficient mice.

The experiment proposed by the reviewer #2 overlooks the fact that pre-pro-B/pro-B cells requires IL-7 signaling to survive. Experiments that use *IL-7R^{-/-}* or *IL-7^{-/-}* mice will have reduced B cell numbers because of the lack of pro-survival signals earlier on in development. While the idea of this experiment attempts to directly address the impact of reduced IL-7 signaling, it is not feasible to uncouple the pro-survival roles from the impact of early *Igk* recombination. Thus, it is unlikely that this experiment will yield a straightforward answer.

Reviewer #4 (Remarks to the Author):

N/A